# Critical angle reflection imaging for quantification of molecular interactions on glass surface

Guangzhong Ma [1], Runli Liang[1,2], Zijian Wan[1,2] & Shaopeng Wang [1✉]

Quantification of molecular interactions on a surface is typically achieved via label-free techniques such as surface plasmon resonance (SPR). The sensitivity of SPR originates from the characteristic that the SPR angle is sensitive to the surface refractive index change. Analogously, in another interfacial optical phenomenon, total internal reflection, the critical angle is also refractive index dependent. Therefore, surface refractive index change can also be quantified by measuring the reflectivity near the critical angle. Based on this concept, we develop a method called critical angle reflection (CAR) imaging to quantify molecular interactions on glass surface. CAR imaging can be performed on SPR imaging setups. Through a side-by-side comparison, we show that CAR is capable of most molecular interaction measurements that SPR performs, including proteins, nucleic acids and cell-based detections. In addition, we show that CAR can detect small molecule bindings and intracellular signals beyond SPR sensing range. CAR exhibits several distinct characteristics, including tunable sensitivity and dynamic range, deeper vertical sensing range, fluorescence compatibility, broader wavelength and polarization of light selection, and glass surface chemistry. We anticipate CAR can expand SPR's capability in small molecule detection, whole cell-based detection, simultaneous fluorescence imaging, and broader conjugation chemistry.

---

[1] Biodesign Center for Biosensors and Bioelectronics, Arizona State University, Tempe, AZ, USA. [2] School of Electrical, Computer and Energy Engineering, Arizona State University, Tempe, AZ, USA. ✉email: Shaopeng.Wang@asu.edu

 1

Molecular interactions are ubiquitous in biological systems and important to the understanding of molecular biology and drug discovery. Surface plasmon resonance (SPR) is the most widely used label-free technique in pharmaceuticals and research laboratories for measuring molecular binding kinetics[1–3]. Owing to the sharp response to the refractive index change on the surface, the high sensitivity of SPR enables the detection of biomolecules[3], small molecules[4], viruses[5], and cells[6,7]. To generate SPR on the surface, the sensor chip (glass slide) must be coated with a metal film (often gold), which is laborious and increases the operation cost. The gold film is not required for glass-based biosensors, such as interferometers[8–10], microring and microsphere resonators[11–13]. However, these sensors are made through microfabrication and still labor-consuming. Reflectometry based on measuring phase shift can directly quantify binding kinetics on a cover glass[14,15], but owing to the instrumentation complexity and moderate sensitivity, it is not as competitive as SPR. It is desirable to develop a simple and sensitive technique for molecular interaction measurements.

The sensitivity of most label-free optical biosensors originates from the response of the probe light to the refractive index change on the sensor surface[16]. For example, in SPR, the reflectivity of the light is significantly reduced at a specific incident angle when the light energy resonates with the surface plasmons in the gold film, and the resonance angle is highly sensitive to the refractive index changes near the surface. By measuring the resonance angle shift, we can measure molecular interactions on the surface that changes the local refractive index. Inspired by this principle, we thought that the molecular interactions induced refractive index changes can also be detected on bare glass when the incident angle is close to the critical angle. Because the critical angle is also refractive index dependent, therefore, the critical angle shift should be a measure of the refractive index change near the glass surface. Indeed, previous studies have shown that $10^{-6}$ refractive index change can be resolved on the glass-water interface by measuring the reflectivity near the critical angle[17]. Detection of particles and hemolysis was also explored with this approach[18,19]. However, this method has never been developed into a molecular interaction detection technique to the best of our knowledge. Although total internal reflection based methods have been used for measuring binding kinetics[8,9,12,15], imaging nanoparticles[20], and obtaining infrared spectra of molecules and cells[21,22], the incident light in these methods is set above the critical angle, where the reflectivity is saturated and the refractive index sensitivity is totally lost.

Taking advantage of the refractive index-dependent nature of the critical angle, we developed a technique called critical angle reflection (CAR) imaging to measure the molecular interactions on a glass surface. CAR presents several unique features compared to SPR. Firstly, the sensitivity of CAR increases with incident angle and can be higher than SPR as the angle approaches the critical angle, allowing CAR to measure small molecules that are challenging for SPR. Also, CAR uses bare cover glass, which is simpler, more robust, and compatible with fluorescence measurement than gold-coated cover glass used by SPR, allowing simultaneous measurement of binding kinetics and fluorescence or total internal reflection fluorescence. Owing to the similarity in detection principle, CAR measurements can be readily implemented in existing SPR setups without the need for extra hardware. We measured the binding of proteins, nucleic acids, and small molecules, and performed cell-based measurements to demonstrate the advantages of CAR on a commercial SPR imaging setup and a home-built objective-based SPR microscope. We anticipate CAR will broaden the capability of SPR with increased sensitivity, concurrent fluorescence imaging, glass surface chemistry, and lower sensor fabrication cost.

## Results

**Detection principle**. The CAR imaging setup is the same as Kretschmann SPR imaging setup[23], where the collimated incident light is reflected by the glass surface and a detection camera is focused at the sample layer on the surface to collect the reflected light (Fig. 1a). The contrast of the image comes from reflectivity changes, which are modulated by the refractive index changes on or near the sensing surface. There are two adjustments in experimental conditions from SPR: (1) the sensor chip is a bare cover glass instead of a gold-coated cover glass; and (2) the incident light is set at slightly below the critical angle, whereas in SPR imaging the incident angle is normally set to slightly below the SPR resonance angle, which is a couple of degrees higher than the critical angle. To perform measurements, the glass surface is functionalized with receptor molecules to capture the ligands in the solution, and upon ligand binding, the refractive index near the surface changes, leading to a change in the reflected light intensity. By measuring the intensity change with a camera, the receptor–ligand interaction can be monitored in real-time.

The detection principle of CAR with p- or s-polarized light can be described by the Fresnel equation with a similar format. We use the equations for p-polarized light below as an example. When a p-polarized light is introduced into a glass prism at an incident angle $\theta_i$ and reflected at the interface between the glass and an aqueous solution, as shown in Fig. 1a, the reflectivity (power reflection coefficient) $R_p$ is given by[24]

$$R_p = \left| \frac{n_g \sqrt{1 - \left(\frac{n_g}{n_a}\sin\theta_i\right)^2} - n_a\cos\theta_i}{n_g \sqrt{1 - \left(\frac{n_g}{n_a}\sin\theta_i\right)^2} + n_a\cos\theta_i} \right|^2 \quad (1)$$

where $n_g$ is the refractive index of glass, and $n_a$ is the refractive index of the aqueous solution. $R_p$ increases with the incident angle, and reaches maximum value of 1 at critical angle $\theta_c$, where

$$\theta_c = \sin^{-1}\left(\frac{n_a}{n_g}\right) \quad (2)$$

Scanning $\theta_i$ from below to above $\theta_c$ shows that $R_p$ increases faster as $\theta_i$ approaches $\theta_c$ and finally reaches total internal reflection at $\theta_c$ (Fig. 1b). The sensitivity of CAR arises from the rapid reflectivity change ($\Delta R_p$) near $\theta_c$ caused by the refractive index change in the aqueous solution near glass surface ($\Delta n_a$) due to molecular binding. Since most molecules have higher refractive index than water, molecular binding event at the glass surface usually increases the effective refractive index of the aqueous solution above glass surface ($\Delta n_a > 0$) and results in right-shift of the curve, which will lower the reflectivity ($\Delta R_p < 0$) if $\theta_i$ is fixed at an angle slightly lower than $\theta_c$. For a given $\Delta n_a$, $|\Delta R_p|$ becomes larger as the $\theta_i$ gets closer to $\theta_c$. The experimental results were verified by simulation (Supplementary Figure 1). This unique feature allows us to tune the sensitivity ($|\Delta R_p|/\Delta n_a$) by changing $\theta_i$ (Fig. 1c). On the contrary, the sensitivity of SPR is fixed for $\theta_i$ in the normal measurement range (Supplementary Figure 2).

To evaluate the performance of CAR as a sensing method, we compared its sensitivity and dynamic range with those of SPR using the same instrument. We used p-polarized light as the incident light in both CAR and SPR for the comparison because SPR can only be generated by p-polarized light. The sensitivity and dynamic range are defined as the absolute value of slope and linear range of $R_p$ vs. $n_a$ plot at given angles (Fig. 1c and Supplementary Figure 2), respectively. The units for dynamic range and sensitivity are RIU (refractive index unit) and $RIU^{-1}$. The results for CAR and SPR are plotted in Fig. 1d. At low angles, CAR presents low sensitivity, but the dynamic range can be at

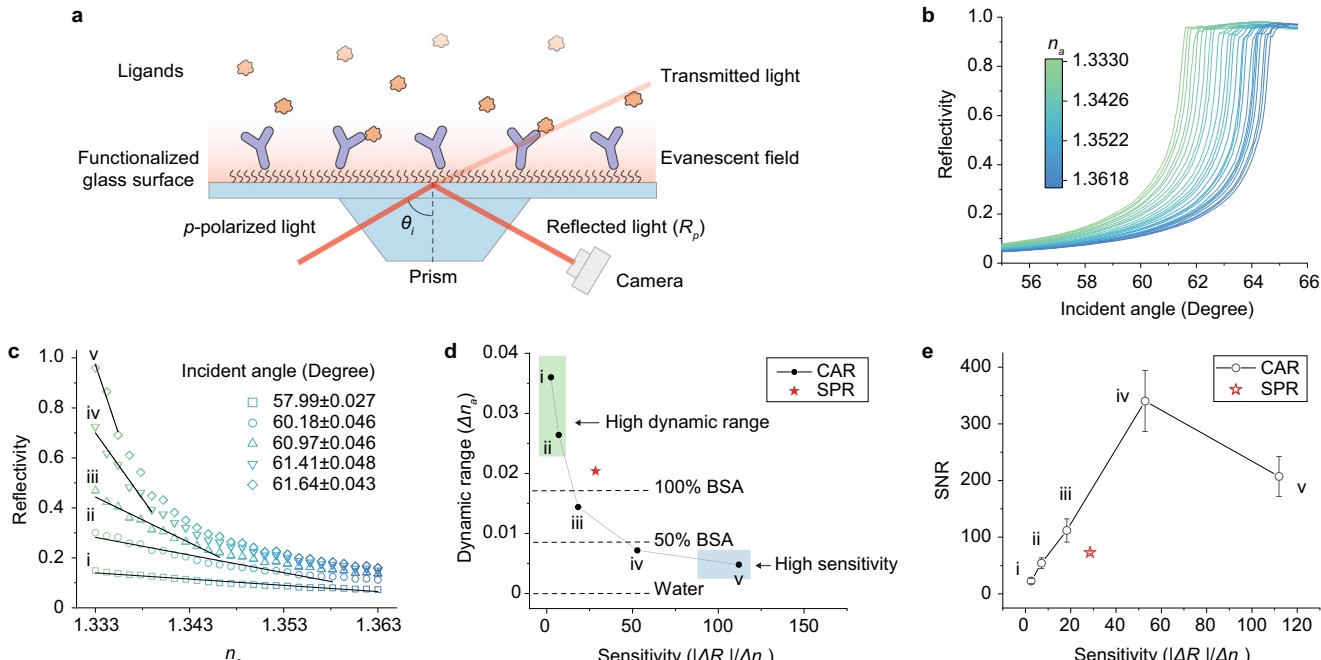

**Fig. 1 Detection principle of CAR. a** Experimental setup and surface chemistry. A protein functionalized cover glass is placed on a prism-coupled SPR imaging setup with index-matching oil. The incident angle $\theta_i$ of a $p$-polarized light is set at slightly below the critical angle. Upon ligands binding to the proteins, the intensity of the reflected light changes and is detected by a camera. **b** Measured reflectivity change as a function of incident angle. The refractive index ($n_a$) on the surface is adjusted by serially adding ethanol to water to make $n_a$ range from 1.3330 (pure water) to 1.3630 (50% ethanol in water). **c** Measured reflectivity as a function of $n_a$ at five representative incident angles (labeled with i to v) with data obtained from **b**. The solid lines show the linear regions (defined by $R^2 > 0.97$, where $R^2$ is the coefficient of determination) of the curves. **d** Tunable sensitivity and dynamic range of CAR. The black dots show the sensitivity and dynamic range determined at the five representative angles in **c**, where the sensitivity and dynamic range are the slope and the range of the linear regions, respectively. The red star marks the sensitivity and dynamic range of SPR, which is not adjustable. To facilitate comparison, the refractive index change induced by the binding of a full layer of BSA (100% BSA), half layer of BSA (50% BSA), and pure water (water) are marked by the dashed lines. The green and blue shadows mark the regions where CAR has high dynamic range and high sensitivity, respectively. **e** Measured signal-to-noise ratio (SNR) of SPR and CAR at the five representative angles. Noise is defined as 1 minute of root mean square of baseline signal. The dots and error bars represent the mean and standard deviation of three measurements.

least two times greater than SPR. At high angles, CAR is five times more sensitive than SPR, but the dynamic range is four times lower. Theoretically, the sensitivity and dynamic range of CAR approaches infinity and zero, respectively, as the incident angle reaching the critical angle, however, the accuracy of the incident angle is limited by the diffraction of the incident light, and thus the exact critical angle is hardly accessible[25,26]. The reduced dynamic range at high angles is still sufficient to measure the binding of medium-sized proteins, for example, bovine serum albumin (BSA, 66 kDa) at up to 25% surface coverage. In between the low and the high angles, CAR has similar sensitivity and dynamic range as SPR. We also measured the signal-to-noise ratio (SNR) of CAR (Fig. 1e). The incident light was set at five representative angles, and 1% ethanol was added to water to generate a refractive index increase (Supplementary Figure 3). The ethanol-induced reflectivity change and baseline fluctuation were defined as the signal and noise, respectively. The maximum SNR of CAR is approximately five times higher than SPR, suggesting CAR is more sensitive to smaller molecules than SPR. Similar sensitivity and dynamic range can be achieved by CAR with $s$-polarization (see Discussion). In addition to the ethanol calibration, which is a standard method in SPR, we also calibrated the sensitivity by coating the sensor surface with a thin polymer layer (Supplementary Figure 4). We found approximately two times difference between the two calibration results, which implies that the Fresnel equation may not be fully accurate when describing reflectivity of a nanometer-scale non-uniform layer. Further investigation of this issue is beyond the scope of this

work, and we use the sensitivity obtained from ethanol calibration in the following sections.

**Biomolecule detection.** To demonstrate the capability of CAR in measuring binding kinetics, we first measured the binding of anti-BSA to BSA, which is often chosen as a model binding pair in SPR (Fig. 2a). BSA was immobilized on the surface of a cover glass (see Methods). Because anti-BSA is a large biomolecule (150 kDa), we tuned CAR sensitivity to medium sensitivity (~25 RIU$^{-1}$, close to SPR) by setting the incident angle at 61.1 degrees. In the experiment, different concentrations of anti-BSA were serially injected over the BSA-coated surface (Fig. 2b). The binding of anti-BSA to BSA increased the refractive index on the sensing surface. After anti-BSA binding in each cycle, the buffer was introduced to the surface to induce the dissociation of anti-BSA from BSA. For ease of comparison with SPR, we converted the refractive index change in CAR (unit: RIU) to resonance unit (RU), a unit typically used in SPR, by 1 RU = $10^{-6}$ RIU (see Methods). By fitting the CAR response curves to first-order binding kinetics, the association rate constant $k_a$, dissociation rate constant $k_d$, and equilibrium constant $K_D$ were determined to be $(1.2 \pm 0.5) \times 10^6$ M$^{-1}$s$^{-1}$, $(1.8 \pm 0.2) \times 10^{-3}$ s$^{-1}$, and 1.5 ± 0.4 nM, respectively. Similar kinetic constants were obtained by measuring the interaction under low and high CAR sensitivities (Supplementary Figure 5), indicating incident angle does not affect the measurement results. To validate the results, we measured the binding pair again with SPR on a gold surface modified with BSA.

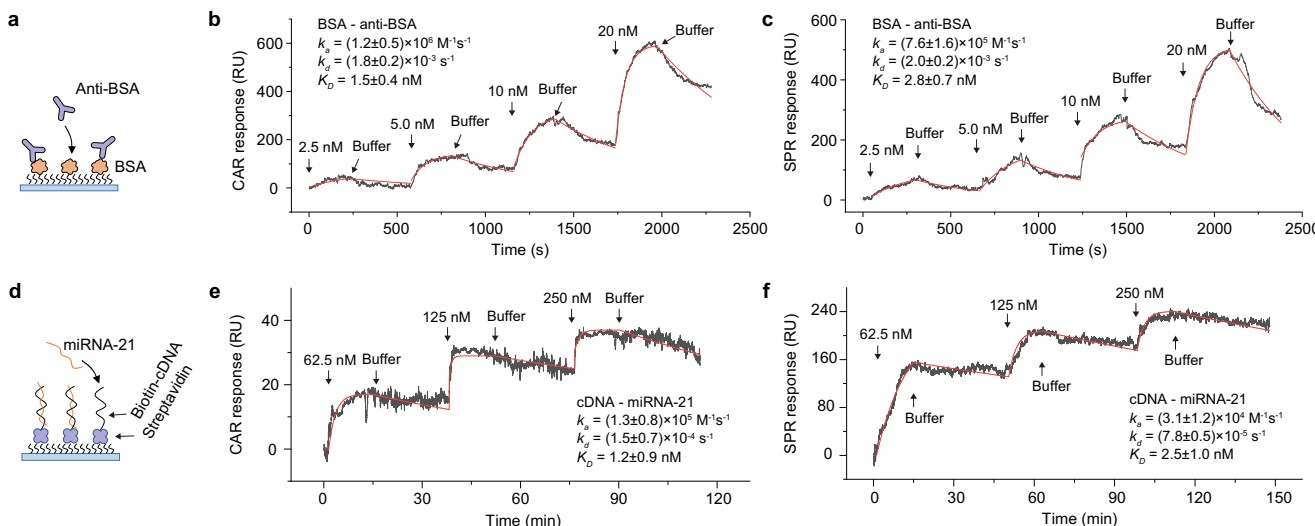

**Fig. 2 Biomolecule detection. a** Bovine serum albumin antibody (anti-BSA) binding to bovine serum albumin (BSA). BSA is immobilized on a glass or gold surface for CAR or SPR measurements. **b** Measuring anti-BSA—BSA binding kinetics with CAR. The incident angle was set at 61.1 degrees with a sensitivity of 25 RIU$^{-1}$. Anti-BSA with different concentrations and buffer were sequentially flowed over the BSA-coated surface. The CAR response (black curve) was fitted to the first order of kinetics (red curve). Note that 1 RU = $10^{-6}$ RIU = 1 pg/mm$^2$ of mass density. **c** Measuring anti-BSA—BSA-binding kinetics with SPR. The experimental conditions were the same as the CAR measurement. **d** MicroRNA-21 (miRNA-21) binding to complementary DNA (cDNA). The biotinylated cDNA was immobilized on a streptavidin-coated glass or gold surface via streptavidin-biotin conjugation. The miRNA-21—cDNA binding was measured with CAR (**e**) and SPR (**f**), and the curves (black) were fitted to the first-order kinetics (red curves). The CAR incident angle was set at 61.4 degrees with a sensitivity of 50 RIU$^{-1}$.

By fitting the binding curves (Fig. 2c), the kinetic constants were determined, with $k_a = (7.6 \pm 1.6) \times 10^5 \text{ M}^{-1}\text{s}^{-1}$, $k_d = (2.0 \pm 0.2) \times 10^{-3} \text{ s}^{-1}$, and $K_D = 2.8 \pm 0.7$ nM. The kinetic constants obtained from CAR and SPR were close, suggesting CAR as an accurate method for binding kinetics measurements.

As an additional example, we measured the binding of a nucleic acid, microRNA-21 (miRNA-21), which is a biomarker for various cancers[27], to its complementary DNA (cDNA). The molecular weight of miRNA-21 is 7 kDa, much smaller than proteins, so we set $\theta_i$ at a higher angle (61.4 degrees) to increase the sensitivity to ~50 RIU$^{-1}$. The glass surface was first modified with streptavidin, and then biotinylated cDNA was immobilized on the surface via biotin–streptavidin conjugation (Fig. 2d). We flowed miRNA-21 and buffer sequentially to the surface to measure the association and dissociation of miRNA-21. The CAR response was recorded, and the kinetic constants were obtained by fitting the response curves (Fig. 2e). The same interaction was also measured with SPR, and the results are shown in Fig. 2f. In principle, the SNR of CAR in the experiment should be several times higher than SPR (Fig. 1e), but the results were not as expected. One reason was because of the difference in cDNA surface coverage on glass and gold. We monitored the immobilization of streptavidin and biotinylated cDNA on glass and gold using CAR and SPR, respectively, and found that the cDNA coverage on gold was 3.6 times as much as that on glass (Supplementary Figure 6). Another reason for the unexpected noise in CAR was that the streptavidin sample had some small aggregates that could not be tightly immobilized on the surface, which were washed off and tumbling around the surface in the following miRNA-21 measurement. This phenomenon was only observed in CAR because CAR is more sensitive to particles in bulk solution (see Discussion).

**Small molecule detection**. At higher incident angle close to the critical angle, the enhanced sensitivity and SNR enable CAR to measure smaller molecules that are challenging for SPR. To address this advantage, we measured the interaction between

carbonic anhydrase II (CAII) and its small molecule ligands: furosemide (331 Da), sulpiride (341 Da), and methylsulfonamide (95 Da) (Fig. 3a). CAII is an enzyme responsible for the catalysis of CO$_2$ hydration, and is found to be related to glaucoma, altitude sickness, obesity, and tumor growth[28]. To perform the measurement, we set $\theta_i$ at 61.6 degrees with a sensitivity of 112 RIU$^{-1}$. CAII was immobilized on a glass surface at 5.8% coverage (Supplementary Figure 7), and the small molecules were flowed over the surface. The binding of each small molecule ligand was measured at several different concentrations and globally fitted to the first-order kinetics. The results are shown in Fig. 3b–d. The small molecules were also measured with SPR on a gold surface with 6.5% CAII coverage (Supplementary Figure 7) using the same experimental conditions, but no obvious signal could be found (Fig. 3e–g). We note that both the glass surface and the gold surface used for this measurement were modified with a monolayer of protein receptors for a fair comparison.

Modifying a three-dimensional matrix such as dextran can further improve the density of the receptors and hence mass change per unit area upon ligand binding. Previous studies[4] show that the same interactions can be measured with SPR using a dextran-coated gold surface, however, the kinetic rate constants were up to 20 times faster than our CAR results. To investigate the discrepancy, we used the same dextran sensor chip and measured the small molecule binding again with SPR. The dextran chip indeed amplified the binding signal. After fitting the kinetics curves, we found that the kinetic rate constants were consistent with our CAR results (Supplementary Figure 8). Next, we checked the diffusion within the sample delivery system, because slow sample diffusion to the sensor surface can distort the binding curve and lead to false slower kinetics. We examined the sample diffusion time by flowing in 1% ethanol solution (Supplementary Figure 9), which ideally should generate a sudden change in reflectivity. In reality, the diffusion time was ~5 s, but still much faster than the time scale of association (~30 s, Fig. 3b–d). Therefore, it is not likely that the kinetics is slowed down by diffusion. Also, by fitting the equilibrium state of the

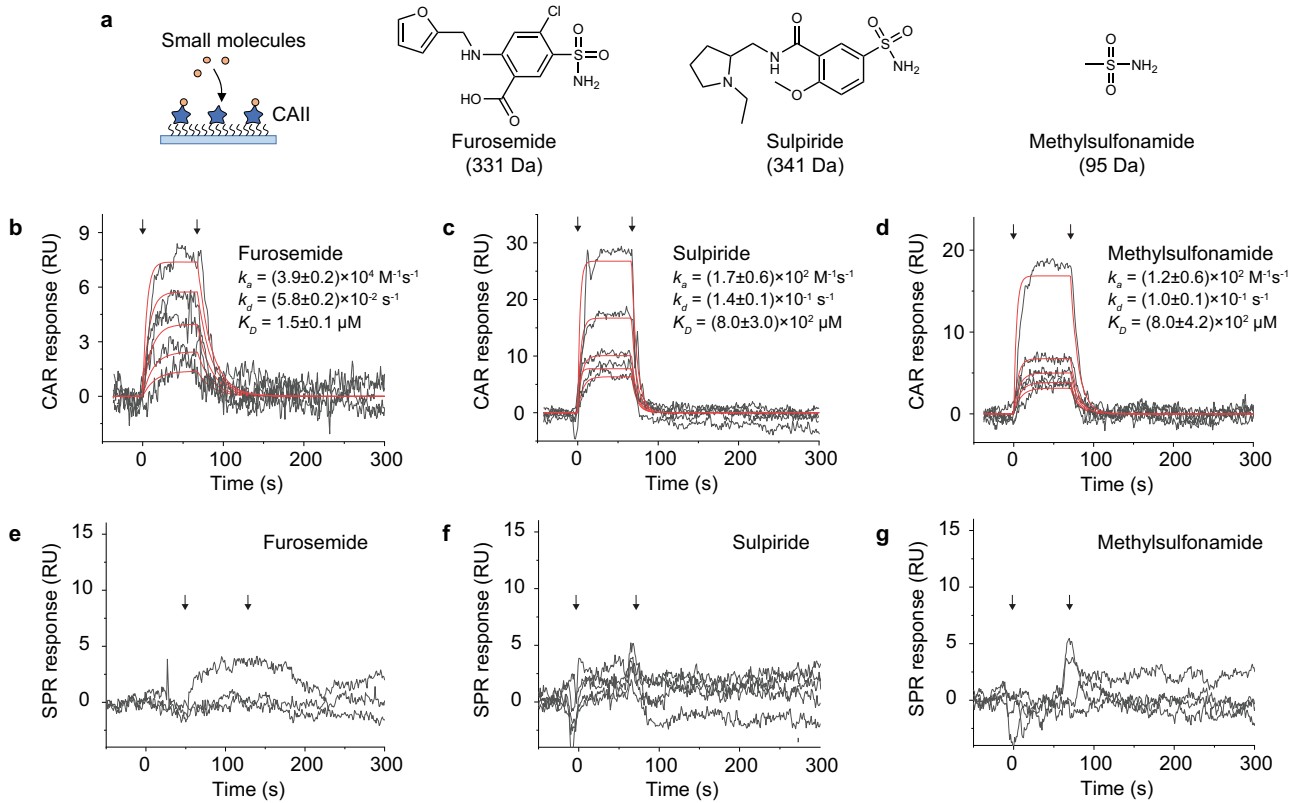

**Fig. 3 Measuring the binding kinetics of small molecule ligands to carbonic anhydrase II (CAII). a** CAII was immobilized on glass for CAR and gold surface for SPR measurements. Three different small molecules, furosemide (331 Da), sulpiride (341 Da), and methylsulfonamide (95 Da), were flowed over the CAII functionalized chip. **b–d** CAR response curves for furosemide, sulpiride, and methylsulfonamide binding (black curves). The incident angle was set at 61.6 degrees with a sensitivity of 112 RIU$^{-1}$. The two arrows mark the starting point of association and dissociation, respectively. The red curves are global fittings of the data to the first-order binding kinetics. Furosemide concentrations: 234 nM, 469 nM, 938 nM, 1.88 μM, and 3.75 μM; Sulpiride concentrations: 62.5 μM, 125 μM, 250 μM, 500 μM, and 1.00 mM; Methylsulfonamide concentrations: 156 μM, 312 μM, 625 μM, 1.25 mM, and 2.50 mM. **e–g** Same interactions were measured with SPR but no clear response was observed. The CAII surface coverages were 6.5% and 5.8% for the gold and the glass surfaces, respectively.

interaction (Supplementary Figure 10), which is not affected by diffusion, $K_D$ is determined to be 1.82 μM, 782 μM, and 1.10 mM for furosemide, sulpiride, and methylsulfonamide, consistent with the real-time values. Based on the above analysis, we conclude that the measured kinetic constants are real, and the discrepancy from the literature value could be due to different CAII protein sources.

**CAR imaging of glycoprotein—lectin interaction on cells**. SPR imaging is known for measuring the binding kinetics between cell membrane protein and ligand directly on the cells without the need of protein extraction and purification[6,29]. Here, we demonstrate that CAR imaging is also capable of cell-based measurements. We studied wheat germ agglutinin (WGA) as an example and measured its interaction with glycoproteins on HeLa cells. WGA is a lectin that can specifically bind to $N$-acet-ylglucosamine structures in the sugar chain of glycoproteins. Investigating the interactions between lectin and glycoprotein is important for understanding the role of glycoprotein in many biological processes, including cell recognition, adhesion, growth, and differentiation[30].

We first used SPR to measure glycoprotein–WGA interaction on fixed HeLa cells (Fig. 4a). The cells were cultured on a gold surface and fixed right before the measurement (Fig. 4b). We flowed phosphate-buffered saline (PBS) buffer over the surface to establish a baseline and then introduced 50 μg/mL WGA (Fig. 4c and Supplementary Movie 1). The binding of WGA to the

glycoproteins increased the surface refractive index and caused the SPR signal to increase. After the association process, PBS buffer was flowed in again to induce dissociation of WGA from the cells. The average SPR response of 10 cells was fitted to the first-order kinetics, and $k_a$, $k_d$, and $K_D$ were determined to be $(2.5 \pm 0.1) \times 10^3$ M$^{-1}$ s$^{-1}$, $(1.3 \pm 0.1) \times 10^{-4}$ s$^{-1}$, and $53 \pm 1$ nM, respectively. The WGA was labeled with Alexa Fluor 488, allowing us to verify the binding using fluorescence. Three fluorescence images were captured at different phases of the binding process: at the baseline, after association, and after dissociation, respectively (Fig. 4d). The fluorescence change, although weak, confirmed that the SPR signal was owing to the binding.

Next, we used CAR imaging to repeat the glycoprotein–WGA binding measurement. The cells were cultured on a glass surface, and the bright-field image and the corresponding CAR image of 9 cells are shown in Fig. 4e. The cells show dark patterns because they have a higher refractive index than the background. $\theta_i$ was set at ~61.0 degrees with similar sensitivity to SPR. Kinetic constants for WGA were determined from the average CAR response of the 9 cells, with $k_a = (6.0 \pm 0.1) \times 10^3$ M$^{-1}$ s$^{-1}$, $k_d = (2.6 \pm 0.1) \times 10^{-4}$ s$^{-1}$, and $K_D = 42 \pm 1$ nM, respectively (Fig. 4f and Supplementary Movie 2). The minor disagreement in kinetic constants might reflect different surfaces and different light-sensing depth between SPR and CAR (more in Discussion). Fluorescence images captured during the CAR measurement confirmed the binding of WGA (Fig. 4g). Notably, the

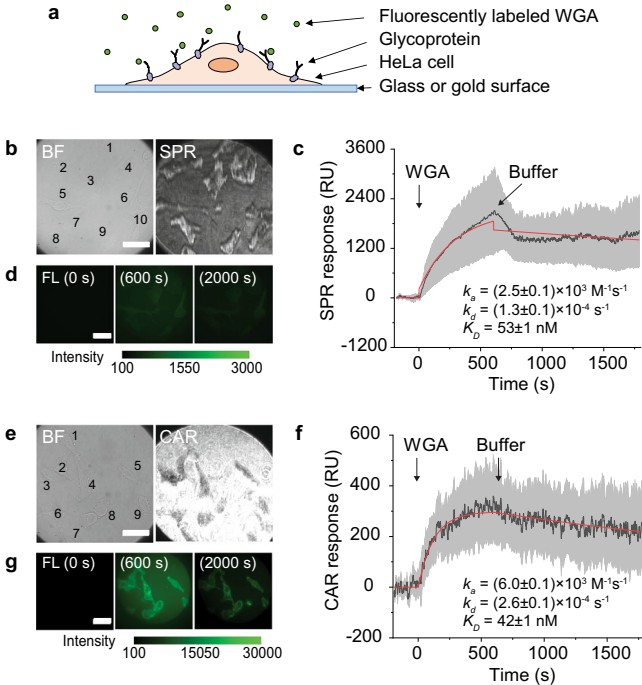

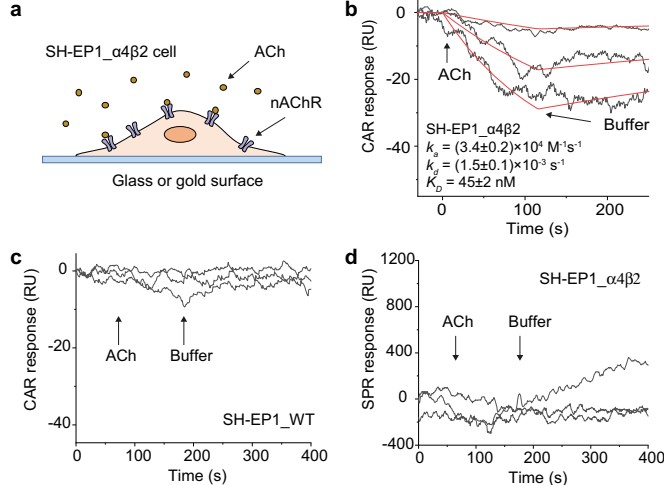

**Fig. 4 Measuring wheat germ agglutinin (WGA) binding to glycoproteins on fixed HeLa cells. a** HeLa cells were grown on a glass or gold surface for CAR or SPR measurements. The cells were fixed with 4% paraldehyde prior to measurements. Alexa Fluor 488-labeled WGA was flowed over the cells and allowed to bind to the glycoproteins on the cell membrane. **b** Bright-field (BF) and SPR images of 10 cells on gold surface. **c** Glycoprotein–WGA binding kinetics measured by SPR. WGA concentration was 50 μg/mL. The black curve and gray shadows are the average SPR signal and standard deviation of the 10 cells (see Supplementary Figure 11a and Supplementary Movie 1 for details), respectively. The red curve is the fitting of the data to the first-order kinetics. **d** Fluorescence (FL) images of the cells captured before WGA binding (0 s), after WGA binding (600 s), and after WGA dissociation (2000 s). Exposure time, 0.1 s. **e** BF and CAR images of 9 cells on the glass surface. **f** Glycoprotein–WGA binding kinetics measured by CAR. WGA concentration was 50 μg/mL. The black curve and gray shadows are the average CAR signal and standard deviation of 8 out of the 9 cells (see Supplementary Figure 11b and Supplementary Movie 2 for details), respectively. The red curve is the fitting of the data to the first-order kinetics. **g** Fluorescence images of the cells captured before WGA binding (0 s), after WGA binding (600 s), and after WGA dissociation (2000 s). Exposure time, 0.1 s. All the scale bars represent 50 μm. The WGA measurements using SPR or CAR were repeated with new sensor chips and cells twice, which showed similar results.

fluorescence intensity on glass is over 30 times stronger than that on the gold surface, which is expected because gold film obstructs light transmission and quenches the fluorescence. For this reason, we believe CAR is more compatible with fluorescence than SPR, and suitable for measuring biological samples that need simultaneous fluorescent labeling.

**CAR imaging of ion channel-small molecule interaction on cells.** Most drugs are small molecules, and over 50% drug targets are membrane proteins[31]. SPR imaging can measure interactions directly on cells, but the sensitivity is inadequate for small molecule ligands. This weakness can be compensated by CAR owing to its tunable sensitivity. To demonstrate this capability, we measured the binding kinetics between acetylcholine (182 Da), a small molecule neurotransmitter, and nicotinic acetylcholine receptor (nAChR), an ion channel membrane protein that is

**Fig. 5 Measuring acetylcholine binding to nicotinic acetylcholine receptor (nAChR) on SH-EP1_α4β2 cells. a** SH-EP1_α4β2 cells were grown on a glass or gold surface for CAR or SPR measurements. The cells were fixed with 4% paraldehyde before the measurement. **b** Acetylcholine (ACh) binding to SH-EP1_α4β2 cells measured by CAR. CAR angle was set at 61.6 degrees with a sensitivity of 112 RIU$^{-1}$. The binding kinetic curves (black) were obtained by averaging the CAR response of seven cells and globally fitted to the first-order kinetics (red) (see Supplementary Figure 12b for the response of individual cells). Acetylcholine concentrations: 25 nM, 100 nM, and 200 nM. **c** Control experiments using wild-type SH-EP1 cells (SH-EP_WT), which have no nAChR. Acetylcholine and PBS buffer flowed to the cells as indicated by the arrows. No clear CAR response was observed (see Supplementary Figure 12d for individual cells). Acetylcholine concentrations: 50 nM, 100 nM, and 200 nM. **d** Measuring acetylcholine binding to SH-EP1_α4β2 cells using SPR. No response was observed due to insufficient sensitivity of SPR (see Supplementary Figure 12f for individual cells). Acetylcholine concentrations: 50 nM, 100 nM, and 200 nM.

responsible for neurotransmission and drug addiction[32]. nAChR was expressed on brain neuroblastoma SH-EP1 cells by transfecting the cells with human α4β2 receptor (SH-EP1_α4β2)[33]. In this experiment, we set $\theta_i$ at high-sensitivity region (61.6 degrees), and flowed acetylcholine solution over the cells (Fig. 5a). The averaged CAR responses of several cells were fitted globally as shown in Fig. 5b. The binding of acetylcholine-induced negative change to the refractive index on the cell membrane. Although the binding of acetylcholine added mass to the surface, the binding also triggered cell membrane deformation and associated mass movement[33,34], which may reduce the effective refractive index on the sensor surface (more in Discussion). The kinetic constants were close to those measured by a cellular membrane deformation detection method[33,34]. To verify that the CAR signal was indeed due to acetylcholine binding, we performed a control experiment using wild-type SH-EP1 cells, which do not have nAChR. The CAR response was negligible (Fig. 5c). The acetylcholine–nAChR interaction was also measured using SH-EP1_α4β2 cells with SPR, which showed no measurable response due to insufficient sensitivity (Fig. 5d).

## Discussion
CAR presents several technical advances compared with SPR in terms of tunable sensitivity and dynamic range, fluorescence capability, and robust glass sensor surface. In this section, we further explore the difference between CAR and SPR and discuss the potential benefits and limitations of CAR.

CAR has a deeper sensing range than SPR, which can be explained by the imaging principles of SPR and CAR. SPR occurs

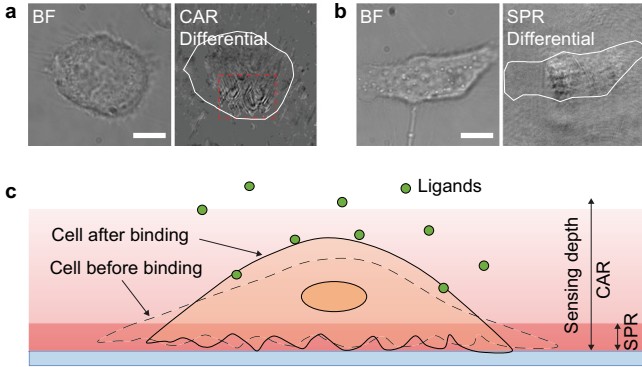

**Fig. 6 Sensing depth of CAR and SPR. a** Bright-field (left) and differential CAR (right) images of a live single cell. The differential CAR image was obtained by subtracting two consecutive frames in an image sequence recorded at 10 frames per second. The white line marks the outline of the cell. The parabolic patterns within the cell (marked by the red square) are generated by the motion of organelles. See Supplementary Movie 3 for details. About 90% of the cells imaged with CAR showed such patterns. **b** Bright-field (left) and differential SPR images of a live single cell. The images were captured and processed under the same condition as in **a**. Organelle motion was not revealed by SPR for the ~20 cells imaged. Scale bars in **a** and **b** represent 10 μm. **c** Schematic picture showing sensing a cell with CAR and SPR. SPR senses the sample with the surface-confined evanescent field and only the bottom section of the sample (several hundred nanometers) can be imaged. CAR has a portion of transmitted light additional to the much deeper evanescent field, which enables CAR to detect the binding-induced deformation of the whole cell.

above the critical angle, and the evanescent field is coupled by the excited surface plasmon, which concentrates the field in the vicinity of the surface (100–200 nm). In CAR, the evanescent field is much less confined to the surface in the absence of surface plasmon. Also, the incident light is below the critical angle, allowing a portion of the light to go beyond the evanescent field range to further distances. An observation in our experiment shows a good example. We found many moving parabolic patterns inside live cells under CAR illumination, which are organelles such as mitochondria (Fig. 6a and Supplementary Movie 3)[35,36]. The parabolic shape arises from the interference between the evanescent wave and scattered light from the organelles[5,35]. However, such patterns did not appear under SPR illumination (Fig. 6b), because the organelles are beyond the detection range of SPR.

The deeper sensing range also reveals cell deformation caused by ligand binding (Fig. 6c)[33,34]. In fact, the CAR signal in WGA binding (Fig. 4f) and acetylcholine binding (Fig. 5b) reflects surface refractive index change caused by a combined effect of bound ligands induced surface mass increase and dynamic mass redistribution owing to binding-induced cell deformation. In both cases, ligand binding increases the surface refractive index because the refractive index of the ligand molecules is higher than the buffer solution, and cell deformation decreases the surface refractive index because the mass center of the cell moves away from the surface. The mass of WGA induces more refractive index change than cell deformation, thus the net signal is positive. Acetylcholine is a small molecule and the signal is dominated by cell deformation, so the net CAR signal is negative.

For molecular interaction studies, the longer sensing depth of CAR could pick up background noises from impurities in the sample, because the motion of particles or aggregates in the sample solution will generate noise to the CAR signal (Fig. 2e and Supplementary Figure 6b) but has minimal impact to SPR.

Another advantage of CAR over SPR is the broader selection of light wavelength. SPR normally uses gold film and incident light with wavelength longer than 600 nm to generate SPR. On the contrary, CAR is compatible with any wavelength in the visible range. In practice, shorter wavelength (such as green/blue light) can be employed to achieve better spatial resolution and shorter penetration depth, which reduces noise from the solution background. UV light also could be used to further improve the spatial resolution and sensitivity, as proteins and nucleic acids absorb lights in the UV range and the signal will be boosted[37]. However, the optics and the camera also need to be UV compatible, and the UV light may cause damage to the sample.

Unlike SPR which requires *p*-polarized light, CAR is not limited by light polarization. Both *p*- and *s*-polarized light present similar sensitivity and dynamic range in CAR (Supplementary Figure 13), indicating CAR measurements can be performed using either or both polarizations at the same time. This capability may allow CAR to measure polarization-sensitive samples and couple with fluorescence anisotropy to determine the orientations and dynamics of molecules.

The incident light in SPR imaging setup may not illuminate the surface at a perfectly uniform angle, which also varies with different instruments. The slight angle difference can barely affect the sensitivity of SPR because SPR has constant sensitivity near the SPR angle (Supplementary Figure 9a). For CAR, however, the sensitivity is strongly dependent on the incident angle, and the imperfect illumination could lead to a non-uniform surface sensitivity (Supplementary Figure 9c). We calibrated the CAR sensitivity of our prism-based setup with 1% ethanol and found that the sensitivity at different regions could differ by up to four times. The spatial sensitivity distribution stems from imperfect collimation of light, because the distribution pattern was independent of different sensor chips and samples (Supplementary Figure 9e). The microscope-based setup showed a more uniform sensitivity because of tunable collimation of incident light and smaller illumination area. For this reason, we always use regions with similar sensitivities when comparing CAR signals in this work.

The detection limit is determined by the noise level and sensitivity. For CAR at high angle, the noise level is $1.7 \times 10^{-4}$ (unit: reflectivity) (Supplementary Figure 3d) and the sensitivity is 112 $RIU^{-1}$ (Fig. 1d). The noise thus corresponds to $1.5 \times 10^{-6}$ RIU, or 1.5 RU, or ~1.5 pg/mm² in mass density[38]. Similarly, the noise for SPR is determined to be 2.4 pg/mm². The sensitivity is intrinsic property of CAR (at a specific angle) and SPR and could not be changed for a given instrument. Therefore, the only way to lower detection limit is to reduce noise. In an ideal scenario, the smallest noise for optical sensors is the shot noise, which is owing to the quantum nature of light. To reach shot noise limit, all other types of noise need to be well under control, such as light source noise and environmental and system mechanical noise. We calculated the theoretical shot noise for our prism-based setup, which is 25 times lower than the measured noise (Supplementary Figure 14). The identified major noise source is mechanical noise from the system cooling fans. Therefore, over an order of magnitude lower detection limit could be reached if the system mechanical noise is reduced with a quiet cooling design.

We have developed a label-free optical sensing method called CAR imaging to quantify molecular binding kinetics on a cover glass surface. CAR measures the reflectivity change near the critical angle in response to molecular binding-induced refractive index changes on the sensor surface. The sensitivity and dynamic range of CAR are tunable by varying the incident angle of light, which allows optimizing the measurement for ligands with different sizes in both biomolecular and cell-based studies. CAR also has a longer vertical sensing range than SPR owing to deeper light

penetration depth. Compared with the gold-coated SPR sensor chips, the glass CAR sensor chips require no surface fabrication and are fully compatible with fluorescence imaging, providing the capability of simultaneous fluorescence imaging. Broader wavelength and polarization selection of CAR may also lead to new applications. Since CAR imaging measurements can be performed on an SPR imaging setup with minimal efforts, we anticipate CAR imaging will become a useful addition to SPR imaging in terms of expanding the capability in small molecule detection, cell-based sensing, and simultaneous fluorescence imaging.

## Methods

**Experimental setup.** The SPR and CAR measurements for principle demonstration (Fig. 1) and protein, miRNA, and small molecule detections (Figs. 2–3) were conducted using a commercial prism-based SPR imaging system (SPRm 200, Biosensing Instrument Inc., Tempe, Arizona) with a 690 nm, 1 mW laser, and a custom installed USB3 CMOS camera (MQ003MG-CM, XIMEA, Germany). The incident angle is controlled via a motor attached to the light source with an accuracy of 5.5 millidegrees and scanning range from 40 to 76 degrees. The system has ×20 magnification. Samples were delivered to the system via an autosampler (BI autosampler, Biosensing Instrument Inc.). The SPR instrument can be directly used for CAR measurement without any modification. The only difference is using a glass sensor chip instead of the gold-coated glass chip and lowering the incident angle from ~70 degrees (SPR angle) to ~61 degrees (critical angle).

All the cell-related experiments including CAR, SPR, transmitted and fluorescence measurements (Figs. 4–6) were performed on an objective-based SPR microscope setup, which consisted of an inverted microscope (Olympus IX-81) and a ×60 (NA 1.49) oil-immersion objective. The light source for CAR and SPR imaging was a superluminescent light-emitting diode (SLED) (SLD-260-HP-TOW-PD-670, Superlum, Ireland) with 670 nm wavelength and 1 mW power set by a SLED current driver (PILOT4-AC, Superlum). The SLED was mounted on a translation stage (PT3, Thorlabs) with a motorized actuator (Z825B, Thorlabs) for adjusting incident angle. The angle accuracy was determined to be ~10 millidegrees. The light source for transmitted and fluorescence imaging were the stocking halogen and mercury lamp of the microscope, respectively. The excitation and emission wavelength used for imaging Alexa Fluor 488 labelled WGA were 488 nm and 518 nm, respectively, and the power density on the sample was ~0.1 mW/cm$^2$. A CMOS camera (ORCA-Flash 4.0, Hamamatsu) was used to record the images. A gravity-based drug perfusion system (SF-77B, Warner Instruments, Connecticut) was used for delivering analytes to the cells.

**Materials.** Cover glass (no.1) for CAR measurements was purchased from VWR. The cover glass was coated with 1.5 nm Cr followed by 43 nm gold using an e-beam evaporator for SPR measurements. Dextran-coated SPR sensor chips were purchased from Biosensing Instrument Inc. (3-glycidyloxypropyl)trimethoxysilane, N-hydroxysulfosuccinimide sodium salt (NHS), N-(3-dimethylaminopropyl)-N′-ethylcarbodiimide hydrochloride (EDC), O-(2-Carboxyethyl)-O′-(2-mercaptoethyl)heptaethylene glycol (SH-PEG8-COOH), BSA, carbonic anhydrase lysozyme II from bovine erythrocytes (CAII), furosemide, sulpiride, methylsulfonamide, and acetylcholine perchlorate were purchased from Sigma-Aldrich. Mouse anti-BSA was purchased from MyBioSource and diluted with PBS 335 times to reach a final concentration of 20 nM. MicroRNA-21 (5′-UAG-CUUAUCAGACUGAUGUUGA-3′) and biotinylated cDNA with A5 spacer (5′ biotin-AAAAATCAACATCAGTCTGATAAGCTA-3′) were purchased from Integrated DNA Technologies. Streptavidin, methyl-PEG$_4$-thiol (MT(PEG)4), and WGA with Alexa Fluor 488 tag were purchased from Thermo Fisher Scientific. PBS was purchased from Corning. Deionized water with a resistivity of 18.2 MΩ·cm was used in all experiments.

**Surface functionalization.** The gold surface was rinsed with ethanol and water for three times and then annealed with hydrogen flame. The cleaned chips were incubated in 0.2 mM SH-PEG8-COOH and 0.2 mM MT(PEG)4 in PBS overnight. Then the -COOH groups were activated by incubating in a mixture of 50 mM NHS and 200 mM EDC for 20 min. In all, 5 μM BSA, 2.2 μM CAII, or 6 μM streptavidin was applied to the surface immediately and incubated for one hour. The remaining activated sites were quenched with 20 mM ethanolamine for 10 min. Finally, the CAII and streptavidin functionalized surfaces were incubated with 1 mg/mL BSA solution to block non-specific binding sites. To immobilize cDNA on the surface, 33 μM biotinylated cDNA was applied to the streptavidin functionalized surface and incubated for one hour.

The glass chip was rinsed with ethanol and water three times. Then the chips were dried with N$_2$, treated with oxygen plasma, and incubated in 1% (3-glycidyloxypropyl) trimethoxysilane in isopropanol overnight. After rinsed with isopropanol and DI water, the chips were immediately incubated with 5 μM BSA,

2.2 μM CAII, or 6 μM streptavidin for 1 hour. Next, 20 mM ethanolamine was used to quench the unreacted sites for 5 min, and 1 mg/mL BSA was applied to the CAII and streptavidin-coated chips for 10 min to block non-specific sites. cDNA was immobilized on the streptavidin-coated surface by incubation in 33 μM biotinylated cDNA solution for one hour.

**Cell culture.** HeLa and SH-EP1 cells were obtained from the American Type Culture Collection, and SH-EP1_α4β2 cells were provided as a gift from Dr. Jie Wu[39]. The cells were cultured in Dulbecco's modified eagle medium (Lonza) with 10% fetal bovine serum (Invitrogen) and 1% penicillin and streptomycin in a humidified incubator at 37°C with 5% CO$_2$. The cells were harvested at 75% confluence, transferred to glass or gold-coated glass chips, and cultured overnight before experiments. The glass and gold surfaces were pretreated with 0.3 mg/mL collagen type IV (Sigma-Aldrich) to improve cell attachment to the surface. For experiments using fixed cells, the cells were fixed with 4% paraformaldehyde solution (Santa Cruz Biotechnology) for 20 min, washed with PBS, and immediately placed on an instrument for measurement.

**Simulation and data processing.** WinSpall (http://res-tec.de/downloads.html, Resonant Technologies GmbH, Germany) was used to simulate the reflectivity as a function of incident angle for CAR and SPR.

For all SPR and CAR detections using the prism-based setup, the images were initially recorded at 100 frames per second and then averaged over every one second by software (ImageAnalysis, Biosensing Instrument Inc.). For SPR and CAR measurements on the microscope-based setup, the images were recorded at 10 frames per second using HCIImageLive software and averaged over every one second. The averaged CAR and SPR images were processed with Fiji[40] to obtain reflectivity change. Then reflectivity was converted to RIU using the ethanol calibration curves (Fig. 1c and Supplementary Figure 2b). The unit RIU was finally converted to a RU by 1 RU = 10$^{-6}$ RIU, which is often used in SPR sensorgrams for result presentation. If needed, RU can be used to estimate mass density or surface coverage of the molecule being measured (1 RU~1 pg/mm$^2$)[38]. After unit conversion, the response curve fitting and binding kinetics constant calculation were carried out with ImageAnalysis and Scrubber (BioLogic Software). All data were plotted using Origin and MATLAB.

**Reporting summary.** Further information on research design is available in the Nature Research Reporting Summary linked to this article.

## Data availability
The data that support the findings of this study are available in Dryad Digital Repository (https://doi.org/10.5061/dryad.47d7wm3d0).

## Code availability
The codes that support the findings of this study are available from the corresponding author upon reasonable request.

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

## Acknowledgements

We acknowledge financial support from the National Institute of General Medical Sciences of the National Institutes of Health under Award Number R01GM124335. We thank Dr. Pengfei Zhang for stimulus discussions. We also thank the reviewers (particularly professor Augusto Garcia-Valenzuela) for constructive review comments.

## Author contributions

S.W. conceived and supervised the project. G.M. designed the experiments. G.M. and R.L. carried out the experiments and analyzed the data. Z.W. fabricated the gold chips. G.M. and S.W. wrote the paper.

## Competing interests

A US provisional patent application (63/122.687) has been filed by Arizona Board of Regents on behalf of Arizona State University based on an early draft of this article. Inventors: S.W., G.M., and R.L.
