## [Peer Review File · Nature Communications]

Reviewers' Comments:

Reviewer #1:

Remarks to the Author:

This paper examines an optical technique referred to as Critical Angle Reflection (CAR) for quantifying molecular interaction near a glass surface. It is based on measuring and imaging changes in the reflectivity in an internal reflection configuration and for angles of incidence very close to the critical angle. Two alternatives are investigated experimentally: One measures the reflectivity at a single angle of incidence with monochromatic light (Laser) and the other one forms an image with quasi-monochromatic light (LED) reflected near the critical angle. The potential to monitor chemical reactions near the glass surface with both variants of the technique are exemplified in this work. Measurements by these techniques are compared, experimentally, with analogous measurements obtained with Surface Plasmon Resonance (SPR) sensing.

I find the comparison of CAR with SPR quite relevant and interesting. SPR techniques are commonly thought as unique and have been used extensively to detect and quantify molecular interaction near a surface. Demonstrating that CAR can perform similar measurements with even higher sensitivity, while being a relatively simpler technique and more versatile, is provoking and energizing to the community.

It would be helpful to provide and discuss details of the image formation and how is the refractive index value encoded in the image. The angle of incidence and reflection of light reaching different pixels in the photodetector matrix of the camera will be different and thus the intensity-to-refractive-index translation or scattering strength is different for different regions of the image.

An important point, that is left ambiguous in the current text and deserves more careful discussion and possibly some amendments, is the idea of depth of sensing in CAR. This is not a simple topic as it is in the SPR technique. In my opinion this is not a drawback of the technique, but it should be analyzed further.

In my view this paper is, overall, an excellent experimental paper. Basically, it explores and demonstrates new capabilities of CAR with interesting examples of molecular binding experiments and compares them experimentally with SPR. However, in my opinion, the paper needs important revisions. It would benefit considerably from addressing in more detail some interesting issues and ambiguities for further analysis.

Next, I list the main points that need clarification or amendments, and possibly, further discussion. Then I continue with some technical questions.

1. The sensing (or illumination) depth of CAR is not a clear concept. In fact, in general it might not be possible to define a sensing depth, as for SPR or other techniques based on evanescent fields interaction. Nevertheless, under some specific assumptions it is correct to say CAR senses the refractive index of the external medium in the vicinity of the glass surface. I believe this is the case when there is a slow refractive index gradient in which the refractive index decreases with depth into the external medium. I think the authors should address this issue more carefully and consider changing the title of the section "Sample illumination depth". As the authors point out, light incident at an angle smaller than the critical angle is transmitted into the external medium. It can be reflected/scattered back by interfaces or structures away from the glass interface and reach the detector contributing to the reflectivity signal. I understand that a detail account and analysis of this issue is out of the scope of the present paper. But some more insight into this problem must be put forward.

2. CAR imaging principle is not explained in the current version of the paper. The schematics of the experimental set up shown in Fig. 1a appears to show a laser beam incident at a single angle of incidence. For imaging, several angles of incidence must be illuminated. In fact, this is the case since a superluminescent LED (and not a laser) is used for imaging. The optics of the imaging must be explained. From watching the videos in the supplemental material, I think that it is the glass interface in contact with the sample that is imaged. But, exactly what is being imaged? Is the modulation of the reflectivity with the subsurface's refractive index that is supposed to be imaged?

But light is reflected at the interface specularly. If the image plane is at the glass surface, then the image is a spatial map of the strength of scattering at the interface? But, this is not related to the modulation of the reflectivity with the subsurface refractive index. It is not clear how is the image formed and how it should be interpreted. This should be addressed in more detail and in a clearer way in the revised version. The following text in Sect. "Spatial sensitivity distribution": "For CAR, however, the sensitivity is strongly dependent on the incident angle, and the imperfect illumination could lead to a non-uniform surface sensitivity (Figure S6c). We calibrated the CAR sensitivity of our prism-based setup with 1% ethanol and found that the sensitivity at different regions could differ by up to 4 times. The microscope-based setup showed a more uniform sensitivity because of tunable collimation of incident light and smaller illumination area. For this reason, we always use regions with similar sensitivities when comparing CAR signals in this work." Appears to indicate that the image is thought to be a map of the specular reflectivity. How is this achieved? It is not clear how is the image calibrated with Ethanol? Where is the image plane then? This is a central part of the paper (the title of the paper refers to CAR imaging) and should be fully clarified. If the image turns out to be a subsurface scattering strength map the interpretation of images and image calibration would be quite different.

3. CAR does not require-polarized light as SPR does. I would guess that p-polarized light is used in this work because existing SPR setups were used to perform the CAR measurements. Is there any other reason why p-polarization was used? Do the authors see any advantage in using different polarizations in CAR?

4. As the angle of incidence approaches the critical angle, the sensitivity of CAR is limited by the divergence angle of the laser beam (due to diffraction). This has been studied in the past by the present reviewer and collaborators (see Refs. iii and iv below). Possibly this limit to the sensitivity should be mentioned in the paper. Also, other uses of CAR can be seen in Refs. i and ii below. Some of my own papers that address CAR measurements

i. R. Márquez-Islas, A. Pérez-Pacheco, L.B. Salazar-Nieva, A. Acevedo-Barrera, E. Medoza-García, A. García-Valenzuela, Optical device and methodology for optical sensing of hemolysis in hypotonic media, *Meas. Sci. & Technol.* 31, 095701, 11pp (2020).

ii. M. Peña-Gomar, Ma. L. González-González, A. García-Valenzuela, J. Antó-Roca, E. Pérez, "Monitoring particle adsorption by laser reflectometry near the critical angle" *Applied Optics* 43 (32), pp. 5963-5970 (2004).

iii. A. García-Valenzuela, Mary Carmen Peña-Gomar, C. Fajardo Lira, Measuring and sensing a complex index of refraction by laser reflection near the critical angle, *Optical Engineering* 41 (7), pp. 1704-1716 (2002).

iv. M. Peña-Gomar and A. García-Valenzuela, Reflectivity of a gaussian beam near the critical angle with absorbing external media, *Applied Optics*, 39 (28), pp. 5131-5137 (2000).

5. I am not sure cost efficiency is truly important when comparing CAR versus SPR. One should refer to a relative cost reduction in the whole instrument which is much smaller than the cost reduction in the chip alone. CAR has several other useful advantages. So, I do not think it is necessary to stress out a cost reduction, specially because in the percentage in cost reduction in the overall instrument might not be that important. This is just a suggestion. Perhaps it can be claimed differently. As long-term costs of operation? I would think it is more important to stress out that fewer things can go wrong with a clean, bare glass surface than with a gold nanofilm of very specific thickness.

6. In my opinion the near independence on wavelength of CAR and its compatibility with fluorescence imaging should be stressed out in the conclusions section. By the way, it would look to me that the true reason that SPR is not compatible with fluorescent measurements is that fluorescent light is obstructed by the gold film to enter the incidence medium, and not necessarily because gold quenches the fluorescence. Whereas the path of fluorescent light towards the detector or camera in CAR is basically a free path. I wonder if the authors agree and if this should be deliberated more carefully in the text.

Some technical queries:

1. The unit "RU" used in many figures and videos, where is it defined? (in the section "detection

limits" and caption of figure 2, it appears as a side comment). Why is it defined like this? What does the acronym stand for? How it can be negative? (see Fig. S9) The authors should elaborate on this definition of unit and provide a detailed explanation in the main text.

2. How is the angle of incidence scanned?

3. What is the accuracy (uncertainty) and repeatability upon setting the angle of incidence? It would appear that is less than 0.05 degrees.

4. What is the bandwidth of the LED used in the imaging setup?

5. I am not completely sure, but I don't think that dynamic range is defined as the linear range of a sensor's response function. (Third line in fourth paragraph of "Results" section.)

6. When stating the In line 10 of the introduction (at the end of the first page), "Reflectometry" is mentioned as a different technique. However, CAR is one of many reflectometry techniques. I suggest adding an extra adjective to reflectometry in this comment to be more specific.

7. In the section "Detection Limits" How is the refractive index noise signal translated to pg/mm²? This is also left unclear when stating the sensitivity adjustment of CAR to monitor the different experiments. Please give some more details of the underlying assumptions and analysis.

8. In Fig.S7 what is the solid line?

9. At the end of caption of Fig.1, "Noise is defined as 1 minute of root mean square of baseline signal". This is not clear to me. The signal and noise should be measured in the same units to compute the SNR.

Sincerely,

Augusto Garcia-Valenzuela

Reviewer #2:

Remarks to the Author:

The manuscript by Ma and co-workers describes a new technique, critical angle reflection (CAR) imaging, a technology that can be implemented on a commercial SPR apparatus but with potentially higher sensitivity and more flexible dynamic range. The technique works by taking advantage of the exceptional sensitivity to refractive index in the immediate range of the critical angle. In addition to providing, in some cases, higher sensitivity, CAR also benefits from a cheaper and more robust substrate (glass vs gold-covered glass). The authors present multiple examples where CAR provides comparable or greater sensitivity than SPR. The authors are also to be commended on being up front about discussing places where new CAR properties can result in unwanted issues, such as higher fluorescent background. Overall, CAR seems like a valuable and accessible new technique that will find many applications, and I enthusiastically recommend publication after a few minor issues are addressed.

Line 67-68, the authors may also wish to point out that glass cover slips are more chemically robust and less prone to scratch damage than gold-covered SPR substrates.

Line 97-107, my biggest issue with this description is that in many places the authors use the word "sensitivity" when they mean "responsivity". Figure 1d and e, as well as this section, clearly show that in many regimes CAR has a higher responsivity than SPR. However, as shown in Figure 2, this higher responsivity does not always translate into higher sensitivity. The authors should replace many of the uses of "sensitivity" with "responsivity".

Line 150, the authors refer to Figure 3f, but likely mean Figure 2f.

Line 221-222, is the main issue quenching, or does the factor of 30 stem more from transmission of the excitation light?

Line 248, the authors should consider a short paragraph to motivate the rest of the discussion.

Line 287-296, the authors are commended in being transparent about some issues with CAR. I have a related question concerning this non-uniformity: is reproducibility an issue with CAR? Will identical samples produce different signals in different experimental acquisitions due to sample to sample and spatial inhomogeneity? How does this compare to SPR? This should be quantified and discussed.

Line 309, the authors refer to Figure S12, but likely mean to refer to S10.

Line 342, the authors should offer more detail on how the commercial SPR instrument was modified so that the experimental setup can be more readily repeated.

Figure 1, in addition to changing sensitivity to responsivity, the authors should also make it more obvious which figures derive from theory and which from experiment.

Reviewer #3:

Remarks to the Author:

S. Wang and co-workers developed critical angle reflection (CAR) imaging to study and quantify molecular interactions. CAR works based on the principle that the critical angle for total internal reflection is refractive index dependent (as dictated by Snell's law) such that the refractive index change during molecular binding can be explored by measuring the reflected light intensity. The authors have used and benchmarked this technique by studying a variety of molecular binding including protein-ligand binding and comparing with surface plasmon resonance (SPR) measurements. Even though the authors claim that the CAR has higher sensitivity as compared to SPR, more experiments and explanations are required (see below for detailed comments) so, I do not recommend publishing this manuscript in the present form.

- 1) In the CAR technique, the reflectivity decreases as the refractive index (RI) of the sample increase as shown in Figure 1c if the angle of incidence is slightly lower than the critical angle, whereas the experimental data showed in the remainder of the manuscript show that CAR response is increasing with the increase in RI, which is because of the post-processing of data where the reflectivity is converted to CAR response. This might be confusing to the general readers, so authors should explain in detail the process of converting reflectivity to CAR response with the help of raw data in supporting information.
- 2) SPR is sensitive to the gold film thickness but the authors show the sensitivity of SPR in Figure 1d with only one thickness. Have the authors tried different thickness to obtain better sensitivity and compare it with CAR?
- 3) The CAR response value should increase with the molecular weight of the molecules used whereas, in Figure 3b-d, this phenomenon is not observed. For example, the CAR response (at the highest concentration) of methylsulfonamide (95 Da) is lower than sulphiride (341 Da), which is expected. However, the CAR response of furosemide (331 Da) is much lower than methylsulfonamide. Can the authors explain this discrepancy? The work from Myszka, D. G (Ref 4 in the manuscript) shows that the SPR response is increasing with the molecular weight of the analyte irrespective of the concentration.
- 4) In addition to point #3, Myszka, D.G has observed SPR response for the methylsulfonamide at 2.5 mM whereas no response was observed from your SPR setup, why is that?
- 5) In Figure 3, the kinetic rate constants were not consistent with the literature value, the authors assigned this discrepancy due to the difference in surface chemistries. But, I think it could be due to the poor dynamic range at 61.6 degrees (angle of incidence) and also the authors need to study more than three/four concentrations for each molecule.
- 6) In order to validate the response from CAR, the authors need to perform at least one molecular binding experiment with different angles of incidence to prove that the K_d value is not essentially changing. This could also explain the discrepancies of K_d values between SPR and CAR.
- 7) In Figure 5d, the authors claim that no measurable response was obtained due to lack of sensitivity, but I believe if the y-axis is rescaled, we could be able to see the response more clearly.
- 8) It is surprising to see that the measured noise is not improving over different integration time in Figure S10, can the authors explain why is that? It is also not clear how system mechanical noise with a quiet cooling design improves the S/N ratio? For this analysis the image sequences were recorded at 500 frames per second, is the same frame rate was used to obtain all the other data?
- 9) Many experimental parameters are missing, for example, acquisition time/frame rate (as noted in point #8) and what is the wavelength and power density used for the fluorescence experiment shown in Figure 4.
- 10) The authors need to proof-read the manuscript carefully because of typos and mismatch of the figure numbers. For example, in the Detection Limit subsection authors referred to Figure S10 as Figure S12. The spelling of methylsulfonamide in Figure 3g needs to be corrected. Moreover, the authors need to change the word "parked" to "set" when describing the angle of incidence throughout the manuscript.

REVIEWER COMMENTS

Reviewer #1 (Remarks to the Author):

This paper examines an optical technique referred to as Critical Angle Reflection (CAR) for quantifying molecular interaction near a glass surface. It is based on measuring and imaging changes in the reflectivity in an internal reflection configuration and for angles of incidence very close to the critical angle. Two alternatives are investigated experimentally: One measures the reflectivity at a single angle of incidence with monochromatic light (Laser) and the other one forms an image with quasi-monochromatic light (LED) reflected near the critical angle. The potential to monitor chemical reactions near the glass surface with both variants of the technique are exemplified in this work. Measurements by these techniques are compared, experimentally, with analogous measurements obtained with Surface Plasmon Resonance (SPR) sensing.

I find the comparison of CAR with SPR quite relevant and interesting. SPR techniques are commonly thought as unique and have been used extensively to detect and quantify molecular interaction near a surface. Demonstrating that CAR can perform similar measurements with even higher sensitivity, while being a relatively simpler technique and more versatile, is provoking and energizing to the community.

It would be helpful to provide and discuss details of the image formation and how is the refractive index value encoded in the image. The angle of incidence and reflection of light reaching different pixels in the photodetector matrix of the camera will be different and thus the intensity-to-refractive-index translation or scattering strength is different for different regions of the image.

An important point, that is left ambiguous in the current text and deserves more careful discussion and possibly some amendments, is the idea of depth of sensing in CAR. This is not a simple topic as it is in the SPR technique. In my opinion this is not a drawback of the technique, but it should be analyzed further.

In my view this paper is, overall, an excellent experimental paper. Basically, it explores and demonstrates new capabilities of CAR with interesting examples of molecular binding experiments and compares them experimentally with SPR. However, in my opinion, the paper needs important revisions. It would benefit considerably from addressing in more detail some interesting issues and ambiguities for further analysis.

Next, I list the main points that need clarification or amendments, and possibly, further discussion. Then I continue with some technical questions.

Response: We thank the reviewer for providing such useful and detailed suggestions. We have carefully thought about questions and listed our responses below.

1. The sensing (or illumination) depth of CAR is not a clear concept. In fact, in general it might not be possible to define a sensing depth, as for SPR or other techniques based on evanescent fields interaction. Nevertheless, under some specific assumptions it is correct to say CAR senses the refractive index of the external medium in the vicinity of the glass surface. I believe this is the case when there is a

slow refractive index gradient in which the refractive index decreases with depth into the external medium. I think the authors should address this issue more carefully and consider changing the title of the section “Sample illumination depth”. As the authors point out, light incident at an angle smaller than the critical angle is transmitted into the external medium. It can be reflected/scattered back by interfaces or structures away from the glass interface and reach the detector contributing to the reflectivity signal. I understand that a detail account and analysis of this issue is out of the scope of the present paper. But some more insight into this problem must be put forward.

Response:

Thanks for the discussion and suggestion. We agree with the reviewer that the sensing depth of CAR is not as well defined as SPR. We also agree that similar to SPR, CAR senses the refractive index of the external medium in the vicinity of the glass surface, including bulk refractive index change of the medium (such as EtOH calibration) and binding induced refractive index changes. When the receptor layer modified on the surface captures the analytes, an increasing refractive index gradient is created near the surface while the refractive index decreases again to the value of the solution beyond the bound protein layer, as the reviewer pointed out. The increasing refractive index near the surface will change the net effective refractive index n_o of the aqueous solution in the Fresnel equation (Eq. 1), resulting in the intensity variation of reflection light received by the detector, namely the sensor response. Our recent single molecule imaging approaches have demonstrated that analytes can be detected by evanescent based approach because they have larger refractive index than water medium (Nature Methods 2020, 17, 1010–1017, and Nature Communications, 2020, 11, 4768).

When the incident angle is smaller than the critical angle, the light reflected/scattered back by the interfaces, structures or impurities in solution will also create sensor output, becoming one noise source of CAR. However, we should point out that even if we set the incident angle to be larger than critical angle, there still exists unknown far field illumination in the TIR (total internal reflection) scheme (Biophys J. 2014 Mar 4; 106(5): 1020–1032). So, as the reviewer pointed out, further study on the far field illumination in TIR scheme is needed in the future for a complete understanding of the phenomena. We have revised the “Sensing distance” section accordingly (page 7).

2. CAR imaging principle is not explained in the current version of the paper. The schematics of the experimental set up shown in Fig. 1a appears to show a laser beam incident at a single angle of incidence. For imaging, several angles of incidence must be illuminated. In fact, this is the case since a superluminescent LED (and not a laser) is used for imaging. The optics of the imaging must be explained. From watching the videos in the supplemental material, I think that it is the glass interface in contact with the sample that is imaged. But, exactly what is being imaged? Is the modulation of the reflectivity with the subsurface’s refractive index that is supposed to be imaged? But light is reflected at the interface specularly. If the image plane is at the glass surface, then the image is a spatial map of the strength of scattering at the interface? But, this is not related to the modulation of the reflectivity with the subsurface refractive index. It is not clear how is the image formed and how it should be interpreted. This should be addressed in more detail and in a clearer way in the revised version. The following text in Sect. “Spatial sensitivity distribution”: “For CAR, however, the sensitivity is strongly dependent on the incident angle, and the imperfect illumination could lead to a non-uniform surface sensitivity (Figure S6c). We calibrated the CAR sensitivity of our prism-based setup with 1% ethanol and found that the

sensitivity at different regions could differ by up to 4 times. The microscope-based setup showed a more uniform sensitivity because of tunable collimation of incident light and smaller illumination area. For this reason, we always use regions with similar sensitivities when comparing CAR signals in this work.”

Appears to indicate that the image is thought to be a map of the specular reflectivity. How is this achieved? It is not clear how is the image calibrated with Ethanol? Where is the image plane then? This is a central part of the paper (the title of the paper refers to CAR imaging) and should be fully clarified. If the image turns out to be a subsurface scattering strength map the interpretation of images and image calibration would be quite different.

Response:

Thanks for the in-depth question and discussion on the CAR imaging principle.

The detection camera for CAR image is set to focus at the sample layer on the glass surface to collect the reflected light, where the incident light is collimated and set at a single angle below but near critical angle. So similar to classic SPR imaging (Anal Bioanal Chem (2004) 379: 328–331), CAR imaging maps the spatial distribution of the intensity of the reflected light. SLED was used in the objective setup to reduce fringe pattern, although it will also introduce small dispersion in illumination angle. The binding induced local refractive index variation above the glass surface will change the intensity of the reflected light to generate CAR image contrast, when incident angle is near critical angle. In other words, what being imaged is the specular reflectivity, but this reflectivity is modulated by the “optical mass” change above glass surface, same as in SPR imaging. Therefore, ethanol can be used to calibrate the sensor output, because it changes the refractive index above the sensor surface, and thus changing the intensity of reflected light to the camera, which is the collection of ensemble signals at different locations. In case of cell imaging, the local refractive index variation created by the cell changes the local reflectivity, which is detected by the camera and generates the CAR image contrast of cells.

We have revised the “Detection principle” section of the manuscript to have a more clear description on the imaging principle (page 2).

3. CAR does not require-polarized light as SPR does. I would guess that p-polarized light is used in this work because existing SPR setups were used to perform the CAR measurements. Is there any other reason why p-polarization was used? Do the authors see any advantage in using different polarizations in CAR?

Response: Yes, the reviewer is right, *p*-polarized light was used in this work because the incident light polarization was fixed in the commercial SPR instrument. Since the focal point of our paper is to demonstrate one can do CAR measurement without modifying the existing SPR instrument, we did not mention *s*-polarized light in the original manuscript. We agree with the reviewer that using *s*-polarized light may have additional benefits. Below shows our simulation and analysis.

The simulation for *s*-polarization was performed in the same way as we did for the *p*-polarization in Figure S1. Same set of parameters were used except light polarization. The results are shown in the figure below (Figure S12). We calculated the dynamic range and sensitivity of CAR with *s*-light and plotted the data together with *p*-light on the same graph for comparison (Figure S12c). It shows that *s*-polarized light is similar to *p*-polarized light in terms of dynamic range and sensitivity.

Since CAR is not limited to polarization, we anticipate that both p and s lights can be used in CAR simultaneously. This capability may allow CAR to measure polarization-sensitive samples and couple with fluorescence anisotropy to determine the orientations and dynamics of molecules.

In our revised manuscript, we have included the simulation and the potential benefits of CAR with s -light illumination (page 8). We also modified the “Detection principle” section to show CAR is not limited to p -light (page 3).

Figure S12. Simulation results of CAR with s -polarized incident light. (a) Relationship between reflectivity and incident angle at different aqueous solution refractive indices (n_a). (b) Reflectivity vs. n_a at five representative incident angles. The black lines are fittings of the linear regions ($R^2 > 0.97$). (c) Sensitivity and dynamic range of CAR with s -polarization (CAR(s)) and p -polarization (CAR(p)) at the five representative angles. The star marks the theoretical sensitivity and dynamic range of SPR. The CAR(p) and SPR data are adopted from Figure S1C.

4. As the angle of incidence approaches the critical angle, the sensitivity of CAR is limited by the divergence angle of the laser beam (due to diffraction). This has been studied in the past by the present reviewer and collaborators (see Refs. iii and iv below). Possibly this limit to the sensitivity should be mentioned in the paper. Also, other uses of CAR can be seen in Refs. i and ii below.

Some of my own papers that address CAR measurements

i. R. Márquez-Islas, A. Pérez-Pacheco, L.B. Salazar-Nieva, A. Acevedo-Barrera, E. Medoza-García, A. García-Valenzuela, Optical device and methodology for optical sensing of hemolysis in hypotonic media, *Meas. Sci. & Technol.* 31, 095701, 11pp (2020).

ii. M. Peña-Gomar, Ma. L. González-González, A. García-Valenzuela, J. Antó-Roca, E. Pérez, “Monitoring particle adsorption by laser reflectometry near the critical angle” *Applied Optics* 43 (32), pp. 5963-5970 (2004).

iii. A. García-Valenzuela, Mary Carmen Peña-Gomar, C. Fajardo Lira, Measuring and sensing a complex index of refraction by laser reflection near the critical angle, *Optical Engineering* 41 (7), pp. 1704-1716 (2002).

iv. M. Peña-Gomar and A. García-Valenzuela, Reflectivity of a gaussian beam near the critical angle with absorbing external media, *Applied Optics*, 39 (28), pp. 5131-5137 (2000).

Response: We agree with the reviewer that the ultimate limitation of sensitivity is diffraction. We also thank the reviewer for providing the useful references. We have mentioned the limitation and included the references in the revised manuscript (Refs. 18, 19, 25, and 26).

5. I am not sure cost efficiency is truly important when comparing CAR versus SPR. One should refer to a relative cost reduction in the whole instrument which is much smaller than the cost reduction in the chip alone. CAR has several other useful advantages. So, I do not think it is necessary to stress out a cost reduction, specially because in the percentage in cost reduction in the overall instrument might not be that important. This is just a suggestion. Perhaps it can be claimed differently. As long-term costs of operation? I would think it is more important to stress out that fewer things can go wrong with a clean, bare glass surface than with a gold nanofilm of very specific thickness.

Response: We agree that cost efficiency may not be as critical as the other advantages of CAR, we followed the reviewer's suggestion and removed the "cost efficiency" paragraph from Discussion. Instead, we mention using glass is simpler and more robust than gold (we made minor changes in abstract, introduction and conclusion sections).

6. In my opinion the near independence on wavelength of CAR and its compatibility with fluorescence imaging should be stressed out in the conclusions section. By the way, it would look to me that the true reason that SPR is not compatible with fluorescent measurements is that fluorescent light is obstructed by the gold film to enter the incidence medium, and not necessarily because gold quenches the fluorescence. Whereas the path of fluorescent light towards the detector or camera in CAR is basically a free path. I wonder if the authors agree and if this should be deliberated more carefully in the text.

Response: We totally agree with the reviewer. We have included wavelength independence of CAR and fluorescence compatibility in the conclusions section (page 9).

We thank the reviewer for pointing out additional reasons for the incompatibility of SPR with fluorescence. It is true that quenching is only one of the factors that decrease the fluorescent signal on gold. In addition, the gold film reflects part of the excitation light and emission light. We have updated the manuscript accordingly (page 6).

Some technical queries:

1. The unit "RU" used in many figures and videos, where is it defined? (in the section "detection limits" and caption of figure 2, it appears as a side comment). Why is it defined like this? What does the acronym stand for? How it can be negative? (see Fig. S9) The authors should elaborate on this definition of unit and provide a detailed explanation in the main text.

Response: We have now defined resonance unit (RU) where it first appears in the main text (page 4, paragraph 2). For ease of comparison with SPR, we converted the refractive index change in CAR (unit: RIU) to resonance unit (RU), a unit typically used in SPR, by $1 \text{ RU} = 10^6 \text{ RIU}$. RU can also be used to

describe surface coverage (mass density) of molecules. We include more detail about unit conversion in “Simulation and data processing” section on page 11.

In the original Figure S9, as well as Figure 5b, the CAR response has negative RU because the response is a combination of binding induced refractive index change (positive RU) and cell deformation (negative RU). The cell deformation is dominant, making the net RU change negative. The explanation can be found in Discussion section (last paragraph on page 7).

2. How is the angle of incidence scanned?

Response: The commercial SPR instrument has a built-in motor that drives the light source, and a software which controls the motor. The position of the motor is translated to angle by the software. The microscope based SPR uses a 3-dimensional translation stage to move the light source and hence the angle. We have included the details in the Methods section (page 10, Experimental setup).

3. What is the accuracy (uncertainty) and repeatability upon setting the angle of incidence? It would appear that is less than 0.05 degrees.

Response: For the commercial SPR instrument, the accuracy is 5.5 mDeg per motor step (the light source is driven by a motor) which is very accurate and repeatable. The microscope setup scans the incident angle through a translation stage which has a step size of 0.2 μm , about 10 mDeg if converted to angle. We also include this information in “Experimental setup” section on page 10.

4. What is the bandwidth of the LED used in the imaging setup?

Response: The bandwidth of the light source is ± 7 nm for the commercial SPR setup (diode laser) and ± 10 nm for the microscope setup (SLED).

5. I am not completely sure, but I don't think that dynamic range is defined as the linear range of a sensor's response function. (Third line in fourth paragraph of “Results” section.)

Response: Dynamic range can refer to the whole working range of the sensor, but in SPR community it usually refers to the linear range, since linear quantitative response can be obtained in this range for kinetic analysis.

6. When stating the In line 10 of the introduction (at the end of the first page), “Reflectometry” is mentioned as a different technique. However, CAR is one of many reflectometry techniques. I suggest adding an extra adjective to reflectometry in this comment to be more specific.

Response: Thanks for the suggestion, we changed “Reflectometry...” to “Reflectometry based on measuring phase shift...” in the revised manuscript.

7. In the section “Detection Limits” How is the refractive index noise signal translated to pg/mm²? This is also left unclear when stating the sensitivity adjustment of CAR to monitor the different experiments. Please give some more details of the underlying assumptions and analysis.

Response: The refractive index unit (RIU or RU, where 1 RU = 10⁶ RIU) is converted to pg/mm² by 1 RU ≈ 1 pg/mm² (Biacore Sensor Surface Handbook BR-1005-71 Edition AB), which is a standard relation in SPR. But strictly speaking this relation is an approximation based on a previous experimental measurement (E. Stenberg et al., J. Colloid Interface Sci. 1991). The experiment measures protein adsorption on a dextran coated gold chip, thus the derived relation should be most accurate for protein on dextran chips. Since this relation is widely accepted by the SPR community regardless of the type of chips or analytes, we did not mention the underlying assumption. In our revised manuscript, we include the Biacore handbook as a reference to notify the audience that there is an approximation in the conversion (page 11, “Simulation and data processing” section).

8. In Fig.S7 what is the solid line?

Response: The solid line is fitting of the data, from which the dissociation constant K_D is determined. We have updated the figure caption.

9. At the end of caption of Fig.1, “Noise is defined as 1 minute of root mean square of baseline signal”. This is not clear to me. The signal and noise should be measured in the same units to compute the SNR.

Response: The root mean square of baseline signal has the same unit as the signal.

Sincerely,

Augusto Garcia-Valenzuela

Reviewer #2 (Remarks to the Author):

The manuscript by Ma and co-workers describes a new technique, critical angle reflection (CAR) imaging, a technology that can be implemented on a commercial SPR apparatus but with potentially higher sensitivity and more flexible dynamic range. The technique works by taking advantage of the exceptional sensitivity to refractive index in the immediate range of the critical angle. In addition to providing, in some cases, higher sensitivity, CAR also benefits from a cheaper and more robust substrate (glass vs gold-covered glass). The authors present multiple examples where CAR provides comparable or

greater sensitivity than SPR. The authors are also to be commended on being up front about discussing places where new CAR properties can result in unwanted issues, such as higher fluorescent background. Overall, CAR seems like a valuable and accessible new technique that will find many applications, and I enthusiastically recommend publication after a few minor issues are addressed.

Line 67-68, the authors may also wish to point out that glass cover slips are more chemically robust and less prone to scratch damage than gold-covered SPR substrates.

Response: We thank the reviewer for the suggestion. We have mentioned glass is simpler and more robust than gold in the updated manuscript (page 2).

Line 97-107, my biggest issue with this description is that in many places the authors use the word “sensitivity” when they mean “responsivity”. Figure 1d and e, as well as this section, clearly show that in many regimes CAR has a higher responsivity than SPR. However, as shown in Figure 2, this higher responsivity does not always translate into higher sensitivity. The authors should replace many of the uses of “sensitivity” with “responsivity”.

Response: To the best of our knowledge, sensitivity is much more often used than responsivity in SPR. The sensitivity of SPR refers to the output (in our paper is reflectivity) to refractive index change caused by analyte binding (J. Homola, Surface Plasmon Resonance Based Sensors (2006), pp51-52).

Responsivity is often used when there are electronic components in the sensor, such as Analog to Digital converters and amplifiers that influence the output signal. One example showing the difference between sensitivity and responsivity is that an amplifier amplifies the output signal but does not change the signal-to-noise ratio. As a result, the responsivity is enhanced due to higher output signal but the sensitivity is unchanged because the signal-to-noise ratio remains the same (<https://www.baslerweb.com/en/sales-support/knowledge-base/frequently-asked-questions/what-is-sensitivity-and-why-are-sensitivity-statements-often-misleading/14987/>). In our SPR or CAR system, the output signal is directed measured without amplification, so we think “sensitivity” is more appropriate.

Line 150, the authors refer to Figure 3f, but likely mean Figure 2f.

Response: The figure should be Figure 2f, we have fixed this problem.

Line 221-222, is the main issue quenching, or does the factor of 30 stem more from transmission of the excitation light?

Response: We thank the reviewer for bringing this question to our attention. Transmission should be the main issue, although quenching exists as well. We have revised the description in the manuscript accordingly (page 6, paragraph 3).

Line 248, the authors should consider a short paragraph to motivate the rest of the discussion.

Response: We included a transition paragraph in the beginning of discussion section.

Line 287-296, the authors are commended in being transparent about some issues with CAR. I have a related question concerning this non-uniformity: is reproducibility an issue with CAR? Will identical samples produce different signals in different experimental acquisitions due to sample to sample and spatial inhomogeneity? How does this compare to SPR? This should be quantified and discussed.

Response: The spatial non-uniformity of CAR image is originated from the instrument, mainly from the non-perfect collimation of light, which generates a spatial distribution of incident angle over the image. Since the optics is fixed, the pattern of spatial non-uniformity is always the same (for both CAR and SPR). To verify the pattern is constant, we repeated the experiment in Figure S6 (Figure S8 in the updated version) using three different chips, imaged the refractive index change induced by 1% ethanol, and calculated the spatial sensitivity distribution for CAR and SPR. The patterns (Figure e below) are close to those we obtained 4 months ago (Figures a and c below).

For a given instrument, the spatial non-uniformity can be quantified using the above method, which later can be used for normalizing signals obtained in different regions. In the future, if the technique is used for multiplexed measurement with an array of analytes printed on the sensor chip (a popular SPR detection format), one should consider the surface spatial non-uniformity and normalize the signal in different regions.

We have included the above discussion in the main text (“Spatial sensitivity distribution” section on page 8) and supporting information (Figure S8).

Line 309, the authors refer to Figure S12, but likely mean to refer to S10.

Response: We have corrected the issue.

Line 342, the authors should offer more detail on how the commercial SPR instrument was modified so that the experimental setup can be more readily repeated.

Response: The commercial instrument can be directly used for CAR measurement without any modification, one only need a cover glass and tune the incident angle before experiment. We included more detail in Methods section (paragraph 1, page 10).

Figure 1, in addition to changing sensitivity to responsivity, the authors should also make it more obvious which figures derive from theory and which from experiment.

Response: All the data in Figure 1 are obtained from experiment. We have clarified this point in the caption of Figure 1 in the revised manuscript.

Reviewer #3 (Remarks to the Author):

S. Wang and co-workers developed critical angle reflection (CAR) imaging to study and quantify molecular interactions. CAR works based on the principle that the critical angle for total internal reflection is refractive index dependent (as dictated by Snell's law) such that the refractive index change during molecular binding can be explored by measuring the reflected light intensity. The authors have used and benchmarked this technique by studying a variety of molecular binding including protein-ligand binding and comparing with surface plasmon resonance (SPR) measurements. Even though the authors claim that the CAR has higher sensitivity as compared to SPR, more experiments and explanations are required (see below for detailed comments) so, I do not recommend publishing this manuscript in the present form.

1) In the CAR technique, the reflectivity decreases as the refractive index (RI) of the sample increase as shown in Figure 1c if the angle of incidence is slightly lower than the critical angle, whereas the experimental data showed in the remainder of the manuscript show that CAR response is increasing with the increase in RI, which is because of the post-processing of data where the reflectivity is converted to CAR response. This might be confusing to the general readers, so authors should explain in detail the process of converting reflectivity to CAR response with the help of raw data in supporting information.

Response: We included more details about unit conversion in "Simulation and data processing" section (page 11).

2) SPR is sensitive to the gold film thickness but the authors show the sensitivity of SPR in Figure 1d with only one thickness. Have the authors tried different thickness to obtain better sensitivity and compare it with CAR?

Response: The thickness of gold film we used is 47 nm where the sensitivity and dynamic range of SPR are maximum. We performed a simulation below to show the SPR response of different thickness, which is consistent with literature (N. F. Murat et al, DOI: 10.1109/SMELEC.2016.7573637).

3) The CAR response value should increase with the molecular weight of the molecules used whereas, in Figure 3b-d, this phenomenon is not observed. For example, the CAR response (at the highest concentration) of methylsulfonamide (95 Da) is lower than sulpiride (341 Da), which is expected. However, the CAR response of furosemide (331 Da) is much lower than methylsulfonamide. Can the authors explain this discrepancy? The work from Myszka, D. G (Ref 4 in the manuscript) shows that the SPR response is increasing with the molecular weight of the analyte irrespective of the concentration.

Response: It is not always true that ligands with higher molecular weight can generate higher response, especially for small molecules. When a small molecule (~100 Da) binds to a much larger protein (~30 kDa), the binding-induced conformation change of the protein alters the local mass distribution within the protein, which may generate more local mass change (or refractive index change) than adding the small molecule alone. Examples showing SPR response not proportional to the mass of small molecules can be found here (ProteOn XPR36 Experimental Design and Application Guide, pp36 and 39).

Although SPR response is not necessarily proportional to mass, both Ref. 4 and the above SPR manual (p36) show methylsulfonamide (95 Da) has lower response than furosemide (331 Da). To check whether the discrepancy is originated from different detection methods (CAR vs SPR), we measured the same small molecules in Fig. 3 with SPR again but using a dextran coated sensor chip (see our response to comment #4). The result is consistent with CAR (Figs. 3b-d), with furosemide (331 Da) having the lowest response. Thus, we think the discrepancy is due to different CAII protein sources rather than detection methods. We revised the manuscript accordingly (page 5 and Figure S7).

4) In addition to point #3, Myszka, D.G has observed SPR response for the methylsulfonamide at 2.5 mM whereas no response was observed from your SPR setup, why is that?

Response: The measurements by Myszka et al are performed using CM5 sensor chips, which have 3D-structured dextran coated surface with high capacity for protein immobilization. Since more CAII

proteins are immobilized on the chip, small molecule binding signal is amplified. In our paper, the sensor surface does not have such dextran coating, so the response is not measurable.

To make fair comparison with Myszka et al, we performed the SPR measurement again with the same dextran coated chips (see the figure below). All the small molecules showed measurable signals, and the magnitude of the signals (in RU) are comparable to those obtained by Myszka et al. However, the noise level of our result is higher than Myszka et al, which is due to instrument difference. Our SPR instrument is designed for imaging purpose (SPRm200, Biosensing Instrument), which does not have a reference flow channel for drift correction as the non-imaging SPR instrument used by Myszka et al (Biacore S51). We believe lower noise can be achieved if a second reference channel is introduced to our SPR/CAR imaging instrument, but that is beyond the scope of the paper. Our data (Figure 3) is sufficient to demonstrate CAR has better signal-to-noise ratio than SPR.

We have included the above analysis and figure in the revised manuscript (paragraph 3 page 5, and Figure S7).

Figure S7. Measuring small molecules binding to CAII on dextran coated gold surface with SPR. (a) CAII was immobilized on the dextran using NHS/EDC chemistry. Furosemide (331 Da), sulpiride (341 Da), and methylsulfonamide (95 Da), were flowed over the CAII functionalized surface. (b-d) SPR sensor response curves (black) and fittings (red) for the three small molecules. Furosemide concentrations: 238 nM, 475 nM, 938 nM, 1.88 μM , 3.75 μM , and 7.50 μM ; Sulpiride concentrations: 62.5 μM , 125 μM , 250 μM , 500 μM , and 1.00 mM; Methylsulfonamide concentrations: 78.0 μM , 156 μM , 312 μM , 625 μM , and 2.50 mM. Note that the noise level of our result is higher than that in reference 4, which is due to instrument difference. Our SPR instrument is designed for imaging purpose, which does not have a reference flow channel for drift correction.

5) In Figure 3, the kinetic rate constants were not consistent with the literature value, the authors assigned this discrepancy due to the difference in surface chemistries. But, I think it could be due to the poor dynamic range at 61.6 degrees (angle of incidence) and also the authors need to study more than three/four concentrations for each molecule.

Response: The dynamic range at 61.6 degrees is 0.005 RIU (Figure 1d) or 5000 RU, much larger than the signal produced by small molecules (30 RU, Figure 3). Thus, the discrepancy is not due to dynamic range. The above SPR results using dextran chips showing similar kinetic constants to Figure 3 serve as additional evidence.

We also added data for more concentrations in Figures 3b and 3d (Figure S9 is updated accordingly as well) to make each molecule in Figure 3 have 5 concentrations.

6) In order to validate the response from CAR, the authors need to perform at least one molecular binding experiment with different angles of incidence to prove that the K_D value is not essentially changing. This could also explain the discrepancies of K_D values between SPR and CAR.

Response: We measured anti-BSA binding to BSA at low incident angle (58.0 degrees, figure (a) below) and high incident angle (61.4 degrees, figure (b)), the kinetic constants are comparable to what we obtained in Figure 2 (which is at medium angle, 61.1 degrees). The figure is included in supporting information.

Figure S4. Measuring anti-BSA binding to BSA at different CAR incident angles. The incident angle was set at (a) low angle (58.0 degrees) and (b) high angle (61.4 degrees).

7) In Figure 5d, the authors claim that no measurable response was obtained due to lack of sensitivity, but I believe if the y-axis is rescaled, we could be able to see the response more clearly.

Response: We zoomed in Figure 5d and presented with color (see below). In general, the responses are irregular.

8) It is surprising to see that the measured noise is not improving over different integration time in Figure S10, can the authors explain why is that? It is also not clear how system mechanical noise with a quiet cooling design improves the S/N ratio? For this analysis the image sequences were recorded at 500 frames per second, is the same frame rate was used to obtain all the other data?

Response: The noise increases after 50 ms integration because system drifting becomes dominant. Once drifting is dominant, longer integration can only amplify the drifting/mechanical noise. The frequency of mechanical noise usually lies in low frequencies. The below figure shows the noise frequency spectrum of our system obtained by performing temporal FFT with recorded images, where the low-frequency (<30 Hz) domain is mainly mechanical noise. Mechanical noise stems from system instability, factors such as cooling system and motors can generate mechanical noise. In our system, the cooling fan is most likely to be the major mechanical noise source.

500 fps was only used for Figure S10. Lower frame rate (10 or 100 fps) was used for the binding kinetics measurements. We included the frame rate for each measurement in “Simulation and data processing” section (page 11).

9) Many experimental parameters are missing, for example, acquisition time/frame rate (as noted in point #8) and what is the wavelength and power density used for the fluorescence experiment shown in Figure 4.

Response: For all SPR and CAR detections using the prism-based setup, the images were initially recorded at 100 frames per second and then averaged over every one second by software. For SPR and CAR measurements on the microscope-based setup, the images were recorded at 10 frames per second and averaged over every one second. The excitation and emission wavelength used for fluorescence imaging of Alexa Fluor 488 labelled WGA were 488 nm and 518 nm, respectively, and the power density was $\sim 0.1 \text{ mW/cm}^2$. We included the above information in the Methods section (page 10, “Experimental setup”).

10) The authors need to proof-read the manuscript carefully because of typos and mismatch of the figure numbers. For example, in the Detection Limit subsection authors referred to Figure S10 as Figure S12. The spelling of methylsulfonamide in Figure 3g needs to be corrected. Moreover, the authors need

to change the word “parked” to “set” when describing the angle of incidence throughout the manuscript.

Response: We have corrected these issues in the revised manuscript.

Reviewers' Comments:

Reviewer #1:

Remarks to the Author:

All my queries were considered by the authors and rebutted or answered quite satisfactorily. Amendments done to the manuscript following the queries are satisfactory. My recommendation is to publish the revised version.

My last suggestion (optional) would be to amend further the discussion regarding Eq. (1) which is valid only for a half-space of uniform refractive index. In general, a thin film or a stratified medium region near the glass surface must be considered. However, I understand it is not the objective of the paper and is out the scope of this work, and that detailed modeling of the reflection coefficient is not strictly necessary to follow the kinetics of the binding processes and images exemplified in this work. However, I do believe that readers would benefit from knowing that relating accurately the optical reflectivity to molecular interactions near the glass surface, in general would require more involved models.

Best regards,

Augusto Garcia-Valenzuela

Reviewer #2:

Remarks to the Author:

All of my issues have been addressed and I recommend publication.

Reviewer #3:

Remarks to the Author:

The authors have addressed all my comments and I believe the revised manuscript should be published.

REVIEWERS' COMMENTS

Reviewer #1 (Remarks to the Author):

All my queries were considered by the authors and rebutted or answered quite satisfactorily. Amendments done to the manuscript following the queries are satisfactory. My recommendation is to publish the revised version.

My last suggestion (optional) would be to amend further the discussion regarding Eq. (1) which is valid only for a half-space of uniform refractive index. In general, a thin film or a stratified medium region near the glass surface must be considered. However, I understand it is not the objective of the paper and is out the scope of this work, and that detailed modeling of the reflection coefficient is not strictly necessary to follow the kinetics of the binding processes and images exemplified in this work. However, I do believe that readers would benefit from knowing that relating accurately the optical reflectivity to molecular interactions near the glass surface, in general would require more involved models.

Best regards,

Augusto Garcia-Valenzuela

Response: We thank the reviewer for the suggestion. We have included a discussion at the end of detection principle section to point out the limitation of Fresnel equation (Eq. 1) in accurately describing a nanometer-scale non-uniform film. We also provided additional experiment and simulation results in Figure S4 to show this limitation.

Reviewer #2 (Remarks to the Author):

All of my issues have been addressed and I recommend publication.

Reviewer #3 (Remarks to the Author):

The authors have addressed all my comments and I believe the revised manuscript should be published.